# QDFlow: **A Python package for physics simulations of quantum dot devices**

**Donovan L. Buterakos**[1,2*]**, Sandesh S. Kalantre**[1,2,4]**, Joshua Ziegler**[2]**,**
**Jacob M Taylor**[1,2,3,5] **and Justyna P. Zwolak** [1,2,3†]

**1** Joint Center for Quantum Information and Computer Science, University of Maryland,
College Park, MD 20742, USA
**2** National Institute of Standards and Technology, Gaithersburg, MD 20899, USA
**3** Department of Physics, University of Maryland, College Park, MD 20742, USA
**4** Department of Physics, Stanford University, Stanford, CA 94305, USA
**5** Axiomatic AI, Inc., Cambridge, MA 02139, USA

* dbuterak@umd.edu , † jpzwolak@nist.gov

## Abstract

**Recent advances in machine learning (ML) have accelerated progress in calibrating and operating quantum dot (QD) devices. However, most ML approaches rely on access to large, representative datasets designed to capture the full spectrum of data quality encountered in practice, with both high- and low-quality data for training, benchmarking, and validation, with labels capturing key features of the device state. Collating such datasets experimentally is challenging due to limited data availability, slow measurement bandwidths, and the labor-intensive nature of labeling. `QDFlow` is an open-source physics simulator for multi-QD arrays that generates realistic synthetic data with ground-truth labels. `QDFlow` combines a self-consistent Thomas-Fermi solver, a dynamic capacitance model, and flexible noise modules to simulate charge stability diagrams and ray-based data closely resembling experiments. With an extensive set of parameters that can be varied and customizable noise models, `QDFlow` supports the creation of large, diverse datasets for ML development, benchmarking, and quantum device research.**

# 1  Introduction

Among the various quantum computing platforms, quantum dots (QDs) stand out for their scalability potential, compact size, and long coherence times [1]. Operating QD devices, however, remains a formidable challenge, with complexity growing rapidly—often exponentially—as the number of qubits increases. Recent advances in integrating machine learning (ML) with quantum device operation have begun to mitigate these difficulties, offering promising automated control and calibration strategies. For example, ML algorithms have been developed for the fabrication [2,3], characterization [4,5], tuning [6–13], and gate virtualization [14,15] of QD devices.

Developing robust ML models requires access to large and diverse datasets representative of the multi-dimensional parameter space typical of QD devices. Crucially, for supervised ML applications, these datasets must also include metadata that identifies key features, such as the global state (i.e., the number of QDs formed), the charge configuration, and the type of transition lines present. Unfortunately, large volumes of high-quality experimental data can be challenging to obtain as companies and research groups often keep such data proprietary [16]. Limited measurement bandwidth in real-world experiments also constrains the efficient exploration of the entire high-dimensional parameter space in a reasonable time. Generating accurate feature labels for publicly available data is a labor-intensive and time-consuming task that can produce subjective and potentially erroneous labels. Physics-based simulations offer a practical solution: they enable the generation of arbitrarily large datasets while providing direct access to the ground-truth charge states, thereby simplifying the labeling process needed for ML training.

Here, we introduce QDFlow, an open-source Python package for simulating QD systems and generating synthetic data tailored for ML training and applications. The core physics engine in QDFlow employs the Thomas-Fermi approximation to numerically solve for the semi-classical charge density $n(x)$ along a one-dimensional (1D) nanowire. While the current state-of-the-art devices are typically realized by confining charges (electrons or holes) within a two-dimensional (2D) heterostructure, the QDs are ultimately formed within quasi-1D channels within those heterostructures, motivating our choice of a 1D model. In practice, the simulated data produced by QDFlow closely resembles that of linear QD arrays in 2D heterostructures. ML models trained on QDFlow-generated data have been shown to generalize effectively to larger 2D QD arrays [14].

There are several open-source QD device simulators that rely on the constant capacitance model, treating the array of QDs and their associated electrostatic gates as nodes in a network of fixed capacitors [17–19]. Additionally, Ref. [19] allows the capacitances to vary with respect to the number of charges $n$ by introducing an $n$-dependent correction to the capacitance matrix. In contrast, in QDFlow the capacitance parameters are physics-informed, obtained directly from the self-consistent Thomas-Fermi solution rather than imposed heuristically. All key physical observables—such as current, charge states, and sensor readouts—are derived from a capacitance model constructed based on the computed charge density $n(x)$. This ensures that the capacitances evolve dynamically with gate voltages, yielding a more realistic description of device behavior. Furthermore, QDFlow allows for modeling regions with low barriers between

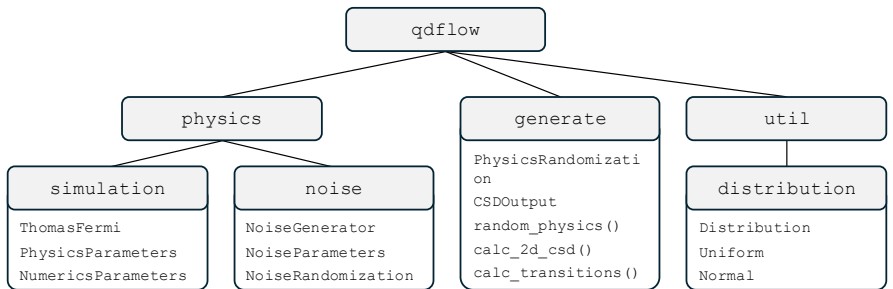

Figure 1: Diagram illustrating the `QDFlow` library organizational structure. Each of QDFlow's four modules is listed, along with the most important classes or functions within those modules.

dots, leading the dots on either side to combine into a single centralized dot. Finally, `QDFlow` incorporates a flexible noise module, enabling the addition of experimentally relevant effects such as thermal broadening, charge offset drift, and voltage fluctuations. These features make the simulated data qualitatively comparable to experimental measurements while maintaining full access to the ground truth labels required for ML applications. Building on `QFlow`—a legacy implementation of the QD simulator that applied the Thomas-Fermi approximation to model charge densities and stability diagrams [20, 21]—`QDFlow` extends these methods into a flexible, open-source framework that integrates physics-informed capacitance modeling with realistic noise processes tailored for ML applications.

`QDFlow` has already demonstrated its utility in advancing ML-driven QD research, confirming that ML models trained exclusively on simulated data can be successfully deployed in experimental settings, on samples fabricated in an academic cleanroom as well as on an industrial 300-mm process line, and on 1D and 2D QD arrays. The legacy version of the simulator was used to generate the `QFlow-lite` dataset [21, 22], which enabled the training of several ML models for global state recognition. The utility of these ML models was demonstrated both offline, by navigating the voltage space within pre-measured experimental datasets [13, 20], and in closed-loop experiments [9, 12, 23]. The dataset also supported the development of a novel classification framework for simple high-dimensional geometrical structures, known as the *ray-based classification* (RBC) framework [24]. The expanded dataset, `QFlow 2.0: Quantum dot data for machine learning` [22], generated using the Thomas-Fermi solver with integrated realistic noise processes, further advanced ML-based approaches to QD tuning. In particular, models trained with data from the `QFlow 2.0` dataset have been successfully applied to tasks such as data quality assessment [25], physics-informed RBC and ray-based navigation in 1D QD arrays [13], and the development of a full virtualization stack for 1D and 2D QD arrays in Ge/SiGe [14] and in Si/SiGe. These successes confirm the compatibility of `QFlow 2.0`-generated with real-world experiments. They also highlight the value of physics-informed synthetic datasets for accelerating the development of automated control tools for QD systems.

`QDFlow` is available for download from the Python Package Index, with the source code released under the GNU General Public License in the `QDFlow` GitHub repository [26]. Comprehensive API documentation is provided via docstrings embedded in the source code and as HTML pages hosted on Read the Docs. The library includes type hints for all classes and functions to support clarity, maintainability, and extensibility. Unit and benchmark tests are also distributed with the `QDFlow` repository to facilitate validation and performance evaluation.

## 2 Physics simulation

QDFlow has three main modules: `simulation`, `noise`, and `generate`, and one utility module, `distribution`, as depicted in Fig. 1. The `simulation` and `noise` modules are part of the `physics` package. The core physics-based engine of the simulator is contained in the `simulation` module. It uses a Thomas-Fermi solver to find the stable charge configuration and sensor output of a particular QD device defined by a set of physical parameters. The QDFlow Thomas-Fermi solver was first introduced in Ref. 21, but has since been refined and extended within QDFlow to improve flexibility, physical relevance, and integration with downstream ML workflows. The `PhysicsParameters` dataclass, which is used to initialize the simulation, specifies over twenty parameters governing the properties of the QD device. These parameters include both material characteristics and device-specific features such as gate geometry and positioning. Importantly, the gate voltages—experimentally relevant control knobs—are explicitly included among the simulation inputs. By sweeping these voltages, QDFlow produces the final outputs: 2D CSDs and 1D rays, directly mirroring the tuning procedures used in real QD experiments.

The `noise` module is responsible for adding noise to the final datasets, as well as for applying certain post-processing to the data. The `generate` module contains high-level functions to assist in generating datasets. It is the module that the user would most often interact with. Finally, the `distribution` module, contained within the `util` package, contains classes defining random variable distributions.

To generate data with QDFlow, the user first chooses whether to run the default configuration or adjust the distributions and ranges over which physics parameters are randomized. Next, they create one or more sets of randomized device parameters, and for each device, generate a CSD using the functions in the `generate` module. Once the physics parameters have been specified, an instance of the `ThomasFermi` class is instantiated. This class serves two main purposes: first, it solves for the charge density function $n(x)$; and second, it uses $n(x)$ to construct a capacitance model and compute physical quantities such as the device's charge state and sensor response. QDFlow then runs the physics simulation for every pixel in each diagram and compiles the results into a `CSDOutput` dataclass, which is returned to the user. By repeating this process over a range of gate voltages, QDFlow generates data that can be assembled into CSDs or rays, depending on the application. The output, stored as `NumPy` arrays, can be directly analyzed and plotted, or optionally augmented with noise to emulate experimental data.

In the following sections, we provide a more detailed account of the nanowire model physics underlying the simulation. We then explain how the Thomas-Fermi approximation is applied to construct the capacitance model that drives the CSD simulation.

### 2.1 Nanowire model

QDFlow employs a 1D physics model in which charges are assumed to be confined to a linear nanowire that lies along the $x$-axis. The ends of the nanowire are connected to electron reservoirs, and a bias voltage can be applied between them. Electrostatic gates are positioned at a height $h$ below the $xy$-plane, and are modeled as infinite cylindrical conductors with central axis parallel to the $y$-axis, as shown in Fig. 2(a). The arrangement makes our nanowire model a hybrid between a true nanowire device and other QD device architectures. Gates biased to low potential act as plunger gates, while those biased to high potential act as barrier gates (for positive charge carriers, with the convention reversed for negative carriers).

The plunger and barrier gates define an electrostatic potential $V(x)$, where $x$ is the distance along the nanowire. Note that because we are using a 1D model, we are only concerned with the potential along the $x$-axis. The potential at a distance $r$ from the center of a single

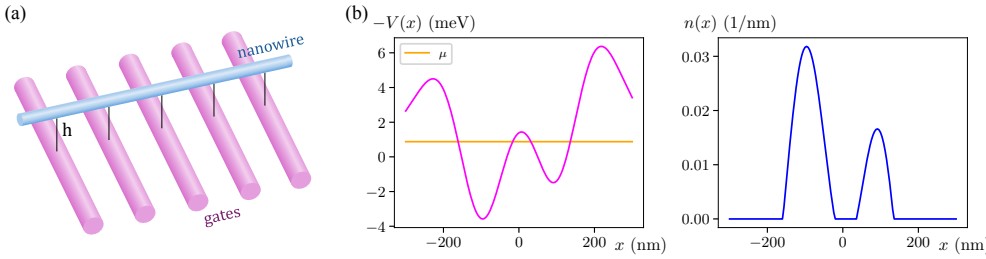

Figure 2: **(a)** The nanowire model used in QDFlow. **(b)** The potential $V(x)$ created by the electrostatic gates (left) and the charge density $n(x)$ induced by the potential (right).

cylindrical gate (in the absence of other gates) can be expressed as the potential of a screened line charge:

$$V_{\text{gate}}(r) = V_h \frac{\mathcal{K}_0(r/\lambda)}{\mathcal{K}_0(h/\lambda)} \tag{1}$$

where $V_h$ is the potential at a reference distance $r = h$ (chosen as the separation between the gate and the nanowire), $\lambda$ is the screening length, and $\mathcal{K}_0(z)$ is the modified Bessel function of the second kind. Specifically, we note that $V_h$ is not the voltage of the gate itself; rather, it is the voltage that the nanowire feels due to the gate (in the absence of other gates). This is essentially the voltage of the gate multiplied by the lever arm of the gate.

Because the presence of nearby gates induces additional charges on each of the gates, and consequently changes Eq. (1), we cannot obtain $V(x)$ by simply summing $V_{\text{gate}}(r)$ over all gates. Determining the exact potential in the presence of multiple gates is a challenging electrostatics problem, even in the purely classical setting. It requires solving the screened Laplace equation with boundary conditions determined by the voltages on each of the gates. While such a calculation can be performed numerically, it must be repeated whenever any gate voltage is changed.

To make the problem tractable, we adopt the simplifying assumption that the induced charges on each gate are rotationally symmetric about the axis of the gate. Under this approximation, the induced charges act as a line charge that lies along the central axis of the gate. Because the gate is rotationally symmetric, the potential $V_{\text{gate}}(r)$ also acts as a single line charge at the center of the gate, and thus the induced charges effectively rescale $V_{\text{gate}}(r)$ by a constant factor. Let $V_i'$ be the rescaled value of $V_h$ for gate $i$ after including the effects of the induced charges on gate $i$, and let $V_i$ be the value of $V_h$ necessary for $V_{\text{gate}}(r)$ to give the actual potential of gate $i$. Then by the superposition principle, $V_i$ can be determined by adding the potential contributions from each of the gates:

$$V_i = \sum_j A_{ij} V_j' \tag{2}$$

where $A_{ij}$ denotes the ratio between the contribution from gate $j$ to the potential at gate $i$ and the effective potential $V_j'$. In the absence of other gates, $V_i = V_i'$, so $A_{ii}$ is simply 1. The contribution from one gate to another is given by $V_{\text{gate}}(r)$, where $r$ is the distance between gates. Thus, $A_{ij}$ can be expressed as follows:

$$A_{ij} = \begin{cases} 1 & \text{if } i = j \\ V_{\text{gate}}(x_j - x_i)/V_{\text{gate}}(\rho_j) & \text{otherwise,} \end{cases} \tag{3}$$

where $x_j$ is the x-coordinate of the central axis of gate $j$ and $\rho_j$ is the radius of the gate $j$. Calculating and inverting the matrix $\mathbf{A}$ allows us to determine the effective potentials $V_i'$ from the applied gate voltages $V_i$. We then obtain $V(x)$ by summing $V_{\text{gate}}\left(\sqrt{(x - x_i)^2 + h^2}\right)$ over all gates and using the effective potentials $V_i'$ in place of $V_h$ in Eq. (1).

## 2.2 Thomas-Fermi solver

Having established how to compute the effective gate potentials and construct $V(x)$, we now turn to the resulting charge distribution along the nanowire. Define $n(x)$ to be the linear charge density at a point $x$ along the nanowire, which is determined in response to the potential $V(x)$, as illustrated in Fig. 2(b). However, due to the Coulomb interaction between charges, the presence of an induced charge in the nanowire will create a correction to $V(x)$. This self-interaction results in the following integral equations, which must be solved self-consistently:

$$n(x) = \int_{\epsilon'(x)}^{\infty} \frac{g_0}{1 + e^{\beta(\epsilon - \mu)}} d\epsilon = \frac{g_0}{\beta} \mathrm{sp}\left[\beta(\mu - \epsilon'(x))\right] \tag{4}$$

$$\epsilon'(x) = qV(x) + \int_{\mathbb{R}} K(x, x')n(x')dx' \tag{5}$$

The physical parameters $\mu$, $g_0$, and $\beta$ in Eq. (4) indicate the Fermi level, the density of states in the conduction band (which is constant in 2D), and the inverse temperature, respectively, and $\mathrm{sp}(z) = \ln(1 + e^z)$ is the softplus function. Parameter $q$ in Eq. (5) controls the sign of the charge carriers, with $-1$ indicating electrons and $+1$ indicating holes, while $K(x, x')$ gives the strength of the Coulomb interaction between points $x$ and $x'$, and is defined as follows:

$$K(x, x') = \frac{K_0}{\sqrt{(x - x')^2 + \sigma^2}} \tag{6}$$

where $K_0$ defines the energy scale of the interaction and $\sigma$ is a softening parameter added to prevent a numerical singularity at $x = x'$, which occurs due to the 1D model breaking down at scales less than the radius of the nanowire. The value of $\sigma$ can be chosen to be $3\pi r/8$, where $r$ is the nanowire radius, to maintain consistency with the potential energy of two uniformly charged disks as the spacing between them approaches zero [27], or alternatively, a custom interaction $K(x, x')$ can be provided.

The Coulomb integral in Eq. (5) is formally taken over the entire nanowire. Because $K(x, x')$ scales for large $x'$ as $1/|x'|$, this introduces concerns that the integral might diverge. At the same time, the integrand is weighted by $n(x')$, which becomes exponentially small for $V(x') - \mu \gg \beta^{-1}$. This condition is satisfied at the external barriers, see Fig. 2(b), provided that the external barrier voltages are sufficiently high. In addition, it is assumed that at distances far away from the nanowire, the system is connected to an electron reservoir where $n(x')$ is large. However, the Coulomb interaction in semiconductors tends to include a screening term that suppresses contributions past a certain range. Thus, in practice, it is sufficient to evaluate the integral between the two external barrier gates.

For convenience, we define a linear operator $\mathbf{K}$ to be the result of evaluating the Coulomb integral as follows:

$$\mathbf{K}f(x) = \int_{\mathbb{R}} K(x, x')f(x')dx' \tag{7}$$

This allows us to combine Eqs. (4) and (5) to obtain:

$$n(x) = \frac{g_0}{\beta} \mathrm{sp}\left[\beta(\mu - qV(x) - \mathbf{K}n(x))\right]. \tag{8}$$

The basic method we employ to solve Eq. (8) is successive iteration. Starting from an initial guess $n_0(x)$, the right-hand side of Eq.(8) is evaluated with $n(x) = n_0(x)$, yielding an updated function $n_1(x)$. This procedure is then repeated until $n(x)$ converges, if at all. The convergence tolerance and the maximum number of allowed iterations are specified through the

NumericsParameters dataclass, which can be provided when instantiating the Thomas-Fermi class. If the iteration does not converge to the specified tolerance within the allowed number of iterations, a ConvergenceWarning is issued.

The convergence can be problematic for certain parameter regimes. For example, if we define $\Delta(x)$ to be the difference between an initial guess $n_0(x)$ and the true value $n(x)$:

$$n_0(x) = n(x) + \Delta(x), \tag{9}$$

then evaluating the right-hand side of Eq. (8) yields:

$$n_1(x) = \frac{g_0}{\beta} \operatorname{sp}\left[\beta(\mu - qV(x) - \mathbf{K}n(x) - \mathbf{K}\Delta(x))\right] \tag{10}$$

We now use the approximation $\operatorname{sp}(z) \approx z$, which is valid for $z \gg 1$. Although this assumption does not always hold (particularly for small $\beta$), it is useful for analyzing certain convergence issues that may arise. Under this approximation, Eq. (10) simplifies to:

$$n_1(x) \approx n(x) - g_0\mathbf{K}\Delta(x) \tag{11}$$

If all eigenvalues of $g_0\mathbf{K}$ are smaller than 1, the error term $-g_0\mathbf{K}\Delta(x)$ will be smaller in magnitude than the initial error $\Delta(x)$, and successive iterations will therefore converge to $n(x)$. Conversely, if $g_0\mathbf{K}$ possesses eigenvalues greater than 1, the iterative scheme will generally diverge. Physically, this divergence corresponds to strong coupling between charges, a regime that is well known to cause convergence difficulties in condensed matter systems [28]. Fortunately, this issue can be partially mitigated by solving Eq. (9) for $\Delta(x)$, substituting the result into Eq. (11), and solving for $n(x)$, yielding the following expression:

$$n(x) \approx (1 + g_0\mathbf{K})^{-1}\left[g_0\mathbf{K}n_0(x) + n_1(x)\right] \tag{12}$$

If we discretize the $x$-axis, the operator $(1 + g_0\mathbf{K})^{-1}$ can be computed through direct matrix inversion. This expression can then be incorporated into the successive iteration scheme by applying Eq. (12) after each iteration. Although there are still parameter regimes where the process diverges, this modified approach drastically enlarges the domain of convergence. In the weak-interaction limit, where the eigenvalues of $g_0\mathbf{K} \ll 1$, the right-hand side of Eq. (12) simplifies to $n_1(x)$ to leading order in $g_0\mathbf{K}$. Thus, in this limit, the method naturally recovers the standard successive iteration procedure.

## 2.3 Capacitance model

After calculating $n(x)$, QDFlow employs a capacitance model to determine the stable charge configuration and other properties. Similar techniques have been implemented in other QD simulations [17–19]. In most of those approaches, the capacitance matrix is assumed to be constant, i.e., the interdot capacitances remain fixed as the gate voltages are swept. The simulation introduced in Ref. 19 allows for variable capacitances by applying a correction to the capacitance matrix based on the particle number. In contrast, QDFlow derives the capacitance matrix directly from the charge density $n(x)$, which depends explicitly on the gate voltages. This feature enables charge-transition slopes and spacings to vary across a single CSD. Moreover, constructing the capacitance model from $n(x)$ naturally captures transitions between a double dot and a merged single dot as the interdot barrier is lowered.

The first step in creating the capacitance model is determining the regions of the nanowire where significant amounts of charge are induced. This is achieved by applying a threshold to $n(x)$, configurable through the NumericsParameters dataclass, and identifying continuous intervals of points that lie above the threshold. This will result in a set of intervals

of the form $[a_i, b_i]$, which we call "charge islands." The thresholding is also responsible for determining whether or not adjacent QDs should be handled as individual dots. Specifically, if $n(x)$ exceeds the threshold throughout the region between the two QDs, they are merged and treated as a single dot. Otherwise, they are considered to be two separate QDs with a potential barrier between them.

Once the charge islands are identified, the energy $E$ of the resulting capacitance model is defined as follows:

$$E = \sum_{i,j} E_{ij}(Q_i - Z_i)(Q_j - Z_j) \tag{13}$$

$$Z_i = \int_{a_i}^{b_i} n(x)dx \tag{14}$$

$$E_{ij} = \frac{1}{Z_i Z_j} \left[ c_k \delta_{ij} \int_{a_i}^{b_i} n(x)^2 dx + \frac{1}{2} \int_{a_i}^{b_i} \int_{a_j}^{b_j} K(x, x')n(x)n(x')dxdx' \right] \tag{15}$$

where $Z_i$ is the (potentially noninteger) charge induced by the gates on island $i$ under the Thomas-Fermi approximation, and $Q_i$ is the integer number of charges on island $i$ under a specific charge configuration $\vec{Q}$. The $c_k$ term of Eq. (15) incorporates the kinetic energy of the charges. Since, for our purposes, the energy matrix fully characterizes the system, we do not compute the capacitances explicitly and instead work directly with the energy matrix. If desired, the capacitance matrix $\mathbf{C}$ can be obtained from the energy matrix via the relationship $\mathbf{C} = (2\mathbf{E})^{-1}$.

After calculating the energy matrix, the next step is to determine the charge configuration $\vec{Q}$ that minimizes the total energy $E$, subject to the constraint that all $Q_i$ must be nonnegative integers. This is an instance of an integer optimization problem, which in general is NP-complete. However, for a moderate number of gates, a brute-force search is sufficient to find the minimum. In particular, we first locate the minimum in the continuous space, which occurs at $\vec{Z}$, and then evaluate $E(\vec{Q})$ over all $\vec{Q}$ such that for each integer $Q_i$, $|Q_i - Z_i| < 1$. Once a stable charge configuration is identified, the potential at each of the sensors is calculated under the assumption that each island $i$ hosts a line of charge with total charge $qQ_i$ and charge density proportional to $n(x)$. The Coulomb potential at each of the sensors arising from these charge islands is calculated, and the result is normalized by dividing by the potential of a single point charge located at a point on the nanowire closest to the sensor in question. This means that a single transition should have a height of no more than 1 after normalization.

Finally, QDFlow allows the current across the nanowire to be found. For this calculation, the left and right sides of the nanowire are assumed to be connected to electron baths with potentials $V_L$ and $V_R$, respectively. The dynamics of the charges are modeled using a semi-classical approach, treating them as particles that travel at the Fermi velocity. Each time they collide with a barrier, the particles have a chance to either tunnel through it or be reflected back. The tunneling probability across each barrier is determined by the transmission coefficient, which we calculate using the WKB approximation. This allows the tunneling rates between islands and the tunneling rates to and from the external charge baths to be obtained. These tunnel rates are then used to define a Markov graph which encodes the dynamics of the transitions between charge states. The current through the nanowire is obtained by evaluating the net rate at which charges enter and leave the charge baths at the steady state of this Markov graph.

## 3  Data generation

The data generation is carried out within the `generate` module. A single instance of the `ThomasFermi` class calculates quantities of interest for a single point in voltage-space only based on the device configuration specified in the `physics` module. To generate a complete CSD, a new simulation instance must be created for each pixel. However, since the gate voltages of neighboring pixels vary only slightly, it follows that the corresponding charge density $n(x)$ will also not change significantly between adjacent pixels. To optimize QDFlow performance, the result of the $n(x)$ calculation at one pixel is used as an initial condition when calculating $n(x)$ at adjacent pixels. This means that $n(x)$ must only be calculated from scratch once for each diagram.

QDFlow contains convenience functions for generating CSDs and rays in the `generate` module. Since the primary purpose of QDFlow is to generate data for training specialized ML models, it is essential that the resulting dataset captures the full range of variability observed in contemporary QD devices. To achieve this, QDFlow includes functionality to randomize nearly all physics parameters and to control the distributions from which each parameter is drawn. This capability is implemented via the `PhysicsRandomization` dataclass, which specifies each physics parameter as either a fixed value (when no randomization is desired, e.g., to allow regeneration of the same QD device), or a `Distribution` from which to draw the random values. A code example in Listing 1 shows how to import QDFlow and how to initialize a random configuration of physical parameters, with $\mu$ drawn from a distribution provided by the user.

The `Distribution`, defined in the `distribution` module, is an abstract base class that encodes how a given parameter should be randomized. Several standard distributions, including `Uniform`, `Normal`, and `LogNormal`, are implemented in QDFlow as wrappers to the NumPy random generator functions of the same name. In addition, user-defined distributions can be easily created by extending the `Distribution` class and implementing the `draw()` method to generate random values in an arbitrary manner. The `CorrelatedDistribution` class handles cases where it is desired or necessary for multiple random variables to be related to one another in some way.

This randomization framework, along with the dozens of configurable physics parameters, enables QDFlow to generate a highly diverse set of CSDs. Figure 3 shows six examples

```python
from qdflow import generate
from qdflow.util import distribution

# Create a new dataclass instance that contains the default
# randomization distributions for each physics parameter
phys_rand = generate.PhysicsRandomization.default()

# Change the range from which mu can be drawn
phys_rand.mu = distribution.Uniform(0, 1.2)

# Generate a list of 6 sets of random device parameters
n_devices = 6
phys_params = generate.random_physics(phys_rand, n_devices)
```

Listing 1: Example code to generate a list of 6 randomized sets of device parameters. First, a `PhysicsRandomization` object is created, which defines the ranges and distributions, as appropriate, from which the physics parameters should be randomized. Distributions for each parameter can then be set as desired.

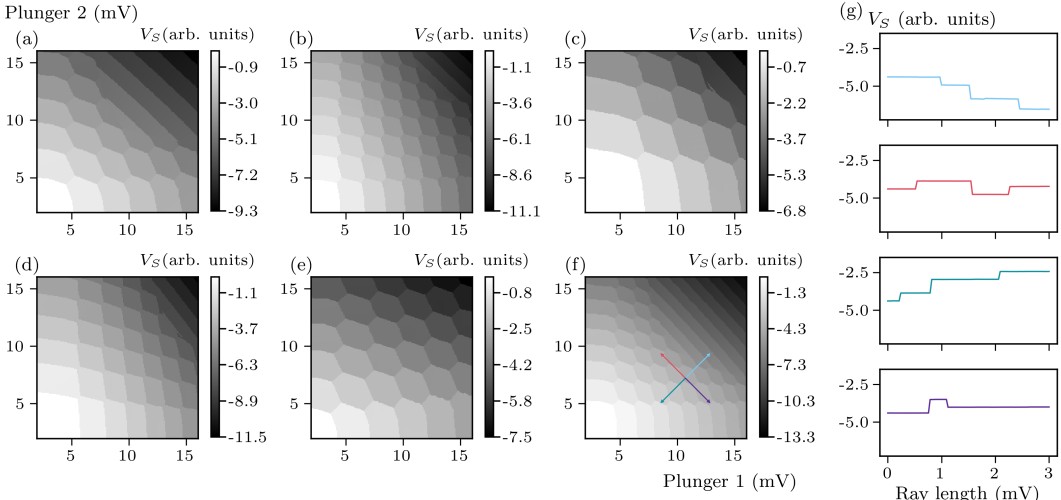

Figure 3: (a)-(f) Examples of CSDs generated with randomized physics parameters. (g) Examples of ray data generated along the rays shown on the CSD in panel (f).

of CSDs generated with `QDFlow`. A snippet of code allowing to generate CSD data from a `PhysicsParameters` object is shown in Listing 2.

While CSDs are extremely useful for visualizing charge states, they require extensive data collection. In practice, only a relatively small subset of points—the charge transitions—are of primary importance. As the number of dots grows, so does the dimensionality of the gate voltage space, rapidly making the exploration of the complete, multidimensional voltage space infeasible. This challenge is typically handled by measuring multiple 2D CSDs, each defined by a different pair of gates.

In Ref. 12, an alternative method for assessing the charge state in QD devices, with 1D rays measured in multiple directions in the voltage space used in place of the 2D CSDs. This method greatly reduces the amount of data required for assessing the charge state of the device, but sacrifices some of the intuitive human interpretability provided by CSDs, necessitating the use of ML tools. To support the development of ML methods for ray-based analysis, `QDFlow`

```python
from qdflow import generate
import numpy as np

# Create a set of physics parameters
phys = generate.default_physics(n_dots=2)

# Set ranges (in mV) and resolution of plunger gate sweeps
V_x = np.linspace(2., 16., 100)
V_y = np.linspace(2., 16., 100)

# Generate a charge stability diagram
csd = generate.calc_2d_csd(phys, V_x, V_y)

# Obtain the sensor readout in the form of a numpy array
sensor_num = 0
sensor_readout = csd.sensor[:, :, sensor_num]
```

Listing 2:   Example code demonstrating how to generate a CSD from a `PhysicsParameters` object.

```python
from qdflow import generate
import numpy as np
from scipy.stats import qmc

# Create a set of physics parameters
phys = generate.default_physics(n_dots=2)

# Generate quasirandom points inside a given area
v_min, v_max = 2., 16.
point_generator = qmc.Halton(d=2, scramble=False)
initial_points = qmc.scale(point_generator.random(n=50), v_min, v_max)

# Define a list of rays that will extend out from each point
ray_length = 3. # length of rays in mV
num_rays = 8
rays = ray_length * np.array([[np.cos(2*np.pi*i/num_rays),
    np.sin(2*np.pi*i/num_rays)] for i in range(num_rays)])

# Generate ray data
resolution = 100 # points per ray
ray_data = generate.calc_rays(phys, initial_points, rays, resolution)
```

Listing 3: Example code demonstrating how to generate ray data from a `PhysicsParameters` object.

includes functionality for generating ray-based datasets, as shown in Listing 3.

# 4 Noise

The simulations described thus far capture many essential physical features of QD devices but omit one critical ingredient: noise. In experimental data, noise strongly influences both the visibility of charge transitions and the reliability of automated analysis.

To more faithfully emulate experimental conditions, QDFlow includes the `noise` module, which contains functionality for adding noise to both CSDs and rays, as well as several postprocessing functions designed to mimic effects of experimental measurements. The module implements several types of noise, including the white noise, pink ($1/f$) noise, telegraph noise, and latching effects, as well as stray transitions arising from nearby unintended dots [25, 29]. Postprocessing functions include adding gate-sensor coupling, adding a sech$^2 x$ blur, and adding Coulomb peak effects. Each noise in QDFlow can be controlled individually, with its magnitude defined relative to the scale of the CSD data, or as a predefined mixture. Similar to the physical parameter randomization discussed in Sec. 3, QDFlow supports designating the distributions from which each noise parameter is drawn via the `NoiseRandomization` dataclass.

Figure 4(a) shows an example of a noiseless CSD simulated with QDFlow. CSDs with noise implementations adapted from QFlow—white, pink, and telegraph noise and Coulomb peak—are depicted in panels (b), (c), (d), and (e), respectively. CSDs with latching, sech$^2$ blur, unintended QD, and sensor-gate coupling—new to QDFlow—are presented in panels (f), (g), (h), and (i), respectively.

The simplest, white noise, is implemented by adding to each pixel a value drawn from a normal distribution with standard deviation given by the magnitude of the white noise. Pink noise is implemented by generating white noise in Fourier space with a uniform random phase

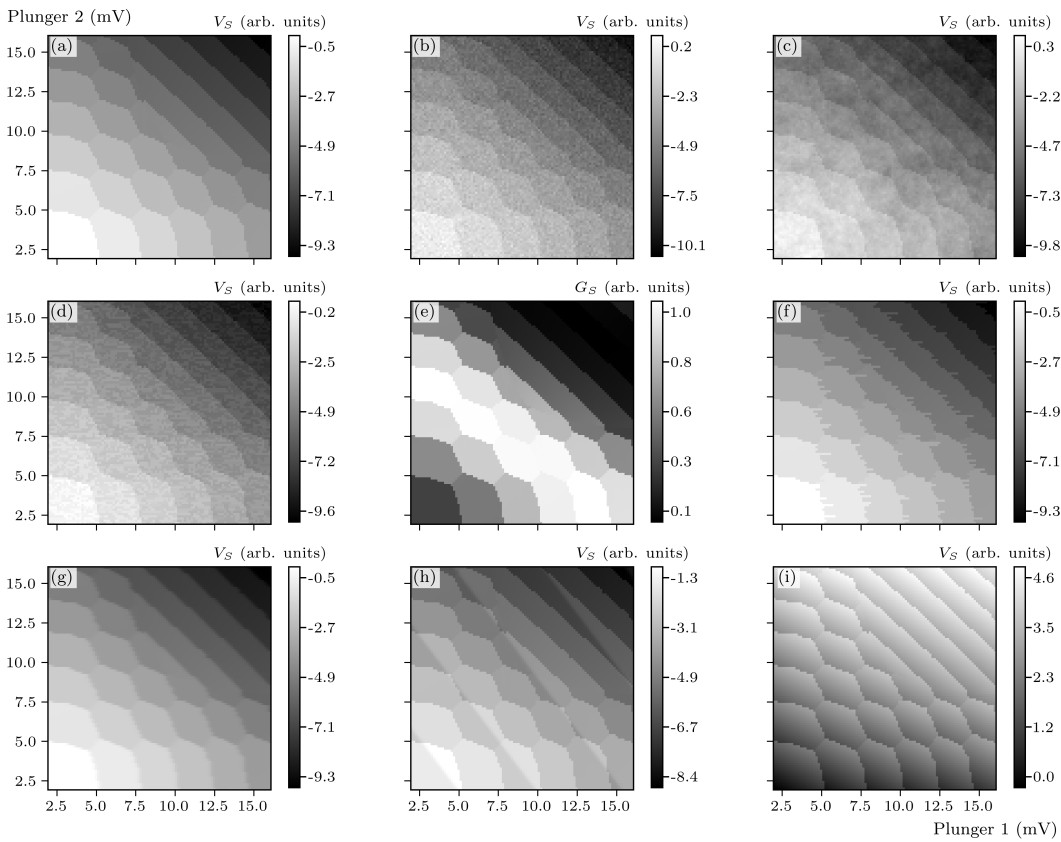

Figure 4: Examples of noise added to a CSD. (a) The original CSD data, (b) white noise, (c) pink noise, (d) telegraph noise, (e) Coulomb peak, (f) latching, (g) Sech blur, (h) unintended QD, and (i) sensor-gate coupling.

and magnitude proportional to $1/\sqrt{f_x^2 + f_y^2}$, where $f_x$ and $f_y$ are the components of each point in Fourier space, and then applying an inverse Fourier transform. This configuration allows for greater variability compared to telegraph noise. Alternatively, QDFlow also provides the option to add pink noise correlated along only the primary measurement axis, which corresponds to the more physical model of pink noise correlated in time.

Telegraph noise is applied along an axis corresponding to the primary measurement direction in experimental data. It consists of adding a value (drawn from a normal distribution with nonzero mean) to a line of several contiguous pixels. The length of the added line is randomly drawn from a geometric distribution. This allows the distribution of lengths of the telegraph noise to follow an exponential, as expected for two-level systems with finite excited-state lifetimes. This process is then repeated across the CSD, alternating the sign of the value added each time.

Transitions from spurious QDs are emulated by adding functions of the form $\tanh((\vec{x}-\vec{x}_0)\cdot\vec{a})$, where $\vec{x}$ gives the coordinates of each pixel, $\vec{x}_0$ is the location on the CSD of the transition, and $\vec{a}$ determines how strongly each of the gates plotted on the $x$- and $y$-axes are coupled to the unintended dot. Values of $\vec{x}_0$ and $\vec{a}$ are randomized, but a single $\vec{a}$ is used if multiple unintended transitions appear on a single diagram.

The latching noise implemented in QDFlow can be controlled using one of two methods. The first and simpler legacy method to simulate latching is to shift each row of pixels by a random number of pixels drawn from a geometric distribution. This process produces a latching-like effect along the charge transitions; however, it is somewhat unrealistic since

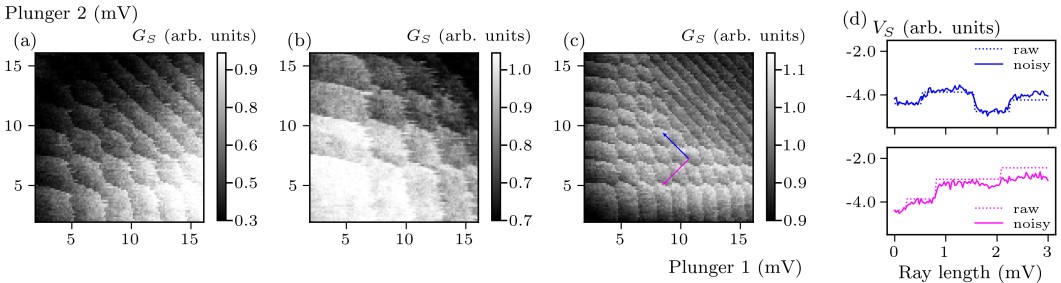

Figure 5: (a)-(c) Examples of CSDs with all noise types combined. (d) Ray data
without noise (dotted line) and with noise added (solid line).

all transitions on the same row are shifted by the same amount, and because pixels far away from transitions are also displaced. A more physically realistic method relies on the nanowire simulation to calculate the sensor readout for both an excited charge state and a stable state at each pixel, similar to what is implemented in Ref. 18. The excited state chosen corresponds to the charge configuration most recently occupied prior to the most recent transition when sweeping the gate voltages. When latching noise is added, the sensor readout from the excited state replaces the stable-state readout for the first few pixels after each transition, with the number of pixels randomized each instance. In general, the second method is preferable as it more accurately reflects the experimental conditions; however, the first, legacy method is provided as a fallback when excited-state data are unavailable or computationally inconvenient, or impossible to obtain.

In addition to noise, several postprocessing functions can be applied to the data. A sensor-gate coupling in the form of a linear gradient along a random direction can be added. Convolving with a $\mathrm{sech}^2 x$ kernel along the measurement axis introduces smoothing of the sharp transitions.

Finally, it is important to note that the physics simulation returns the value of the potential at each of the sensors; however, experimentally, the conductance of the sensor is the quantity that is measured. Therefore, we convert from potential to conductance by using a Coulomb peak lineshape of the form $G \propto \mathrm{sech}^2[A(V - V_0)]$, where $G$ is the conductance of the sensor, $A$ is a parameter that determines the width of the Coulomb peak, $V$ is the potential at the sensor (the simulation output), and $V_0$ is the peak center [30].

CSDs with a mixture of noises optimized for compatibility with experimental data are presented in Fig. 5(a)-(c). Figure 5(d) shows two rays with added noise. The exact amounts of each type of noise are randomized for each diagram. A code example showing how to generate noisy CSD data for a previously simulated sample CSD is shown in Listing 4.

Given the computational complexity of Thomas-Fermi calculations, the `noise` module is configured to assume that a complete noiseless CSD (or a ray-based data) has already been generated using the `generate` module. This approach gives us several advantages. First, it significantly reduces the computational overhead since multiple noise realizations with different relative noise strengths can be generated from a single noiseless CSD. Secondly, the modular approach adopted in QDFlow provides flexibility that cannot be achieved experimentally, where it is not possible to calibrate individual noise sources to the desired level. Additionally, it allows us to specify the magnitudes of each of the noise types relative to the local scale of the surrounding data points, which is important since the scale of the data points can vary across large CSDs. Finally, it allows for systematic benchmarking of ML algorithms since the noise level and type can be modified independently of the underlying physical configuration. For example, one can generate a single noiseless dataset and then apply different realizations of noise to study the robustness of a given algorithm under varying experimental conditions,

```
from qdflow.physics import noise
from qdflow.util import distribution
import numpy as np

# Use data from previous example
data = np.load("sensor_readout.npy")

# Create a new dataclass instance that contains the default
# randomization distributions for each noise type
noise_rand = noise.NoiseRandomization.default()

# How much noise to add, relative to the transition height upper bound
noise_amount = 0.15

# Use a CorrelatedDistribution to randomize the white, pink, and
# telegraph noise so that the total always equals noise_amount
num_dists = 3
dists = distribution.SphericallyCorrelated(num_dists,
        noise_amount).dependent_distributions()
noise_rand.white_noise_magnitude = dists[0].abs()
noise_rand.pink_noise_magnitude = dists[1].abs()
noise_rand.telegraph_magnitude = dists[2].abs()

# Generate a random set of noise parameters
noise_params = noise.random_noise_params(noise_rand)

# Add noise to the data
noisy_data = noise.NoiseGenerator(noise_params).calc_noisy_map(data)
```

Listing 4: Example code to add noise to a charge stability diagram. First, a `NoiseRandomization` object is created, which defines the ranges and distributions, as appropriate, from which the noise parameters should be randomized. A `CorrelatedDistribution` is used to randomize the different noise types while ensuring that the total noise amount is constant.

as was done in Ref. 13.

## 5  Benchmarking and limitations

To benchmark QDFlow, we find the mean runtime of the physics simulation for 10,000 randomly generated sets of physics parameters. Each simulation is run with at most 1,000 iterations and a relative convergence tolerance of $10^{-3}$. We are interested in comparing the contribution to the overall runtime from calculating $n(x)$ with the remaining portion of the simulation as the number of QDs increases. For each number of dots $N$, we obtain two runtimes: (i) the time required to compute the charge density profile $n(x)$, and (ii) the remaining time to construct the capacitance model, minimize Eq. (13), and evaluate the sensor output. The results for small- and mid-sized QD arrays are shown in Fig. 6(a) and Fig. 6(b), respectively.

For a small number of dots ($N \leq 20$), the main bottleneck is the initial part of the simulation where $n(x)$ is calculated. However, as the number of dots $N$ is increased, the minimization of Eq. (13) over integer charge configurations $\vec{Q} \in \mathbb{Z}^N$ for which the runtime scales exponentially in $N$ begins to dominate. In practice, we expect most use cases to involve no more than ten

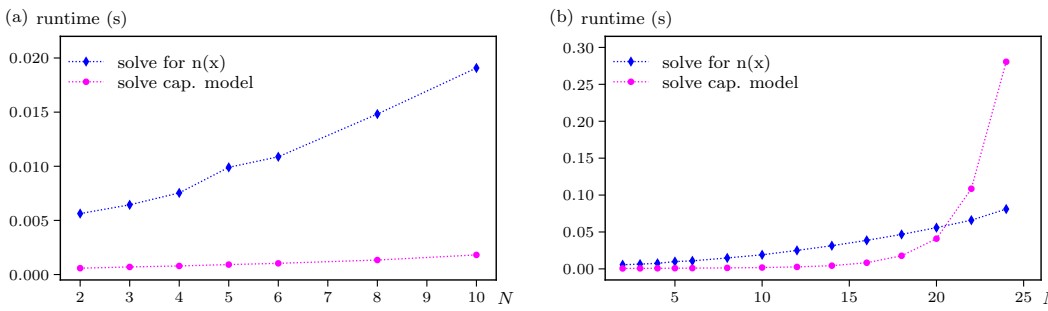

Figure 6: Average runtime versus number of dots $N$ for (a) Thomas-Fermi solver and (b) capacitance model solver. Benchmarks were performed on a 2.8 GHz AMD Ryzen 5 7520U processor.

QDs, as long-range interactions tend to be negligible due to screening from the material and the nearby gates; thus, we focus attention on the cost of computing $n(x)$ in this regime.

Because QDFlow calculates the capacitance model at each point in the CSD by explicitly solving for $n(x)$, it requires several orders of magnitude more runtime to generate CSDs than simulations based on constant capacitance models. Thus, for applications that require real-time data, faster simulations such as QDsim [17] or QArray [18] may be more suitable. On the other hand, for applications such as generating datasets for training and validation of ML models for autonomous control of QD systems, the higher-fidelity data provided by QDFlow—including effects such as QDs merging across low barriers—often justifies the additional computational cost.

In addition to its relatively long runtime, QDFlow has several modeling limitations. While QDFlow is based on an underlying physical nanowire model, it is designed only to produce qualitatively realistic behavior. For example, gates are modeled as infinite cylinders arranged in a simple, idealized geometry, whereas in real devices, the shapes and arrangements of gates can be considerably more complex. Furthermore, the electron density $n(x)$ is calculated only in 1D, an approximation that can introduce errors for systems with non-negligible transverse extent. Despite being a 1D simulation, data generated with QDFlow is qualitatively similar to that from 2D QD arrays, provided that crosstalk is not too large. Influence from other dots can be included either during the simulation phase by extending the nanowire to include additional dots on the left and right sides, or during the noise-adding phase as unintended dot noise. However, the ability to model crosstalk from other dots remains somewhat limited, as QDFlow cannot reproduce data features from 2D architectures with tight couplings between nearby dots. Finally, the simulation uses a semiclassical approach and therefore does not include certain quantum effects, such as finite tunnel coupling effects and charge-state hybridization, which would be captured by more complex models such as the Hubbard model. Taken together, these approximations prevent QDFlow from generating quantitatively accurate data.

QDFlow also lacks certain capabilities present in other simulators. For example, it cannot currently model closed systems with a fixed number of charges isolated from the source and drain contacts—a functionality available in QArray [18]. Additionally, QDFlow does not currently model quantum effects due to finite tunnel coupling, which can cause broadening of the interdot transitions and rounding of sharp corners around the triple-points due to hybridization of charge states. For applications involving virtualization of gates, quantum effects are generally small enough that a semiclassical approximation is sufficient. However, for applications such as readout, where fine control of gates is needed, quantum effects are more important. If these features are relevant, QDarts contains a simulation of the finite tunnel coupling effect [19]. Finally, QDFlow is configured to model the state that minimizes

the energy of the capacitance model, implicitly assuming that the system can transition freely between charge configurations. However, this assumption can break down for large arrays of QDs. For example, if a charge must transition between non-neighboring QDs passing through an intermediate QD that forms a large barrier, it may get "stuck" on one side even if it is energetically favorable to move to the other side. This will result in much more extreme latching effects than QDFlow can model.

QDFlow has been tested both modularly and holistically, through extensive unit testing and by comparing its output to experimental data. The QDFlow repository includes unit tests for every function in the package, ensuring coverage of all lines of code. The validity of QDFlow has been confirmed by performance of multiple specialized ML models trained on QDFlow-generated data and deployed in real-world applications, as we discuss in Sec. 1 and in Sec. 6.

# 6   Conclusion

Progress toward scalable quantum information technologies based on QD systems depends critically on overcoming the complexity of device operation and calibration with increasing number of QDs. Novel ML-based methods have emerged as powerful tools to address these challenges, but their effectiveness relies on access to large, diverse, and accurately labeled datasets. QDFlow was developed precisely to meet this need.

QDFlow differs from existing QD simulations in that it fully simulates the charge density function $n(x)$. By integrating a self-consistent Thomas-Fermi solver with a dynamic capacitance model, QDFlow provides a physics-informed simulation framework that goes beyond constant-capacitance approximations. This enables the generation of CSDs and ray-based data with features that evolve naturally with gate voltages, mimicking experimental behavior such as dot merging and transition slope variation.

The modular data generation tools allow for extensive randomization over physical parameters, yielding highly diverse synthetic datasets suitable for ML applications, while the noise module introduces experimentally relevant effects—including thermal broadening, telegraph noise, latching, and unintended transitions—in a controllable fashion. Together, these features make QDFlow uniquely positioned to support both the development and benchmarking of ML algorithms implemented in a wide range of tuning procedures, device architectures, and material platforms. Early use cases, such as the QFlow-lite and QFlow 2.0 datasets, have already demonstrated QDFlow's ability to accelerate the training of ML models for global state recognition [9,20], ray-based navigation and charge tuning [12,13], data quality assessment [25], detection of spurious QDs [31], and virtualization of QD arrays [13, 14]. As an open-source, extensible platform with comprehensive documentation, QDFlow is designed to serve as both a research tool and a community resource.

QDFlow represents a paradigm shift among QD simulators. Whereas other simulations typically rely on constant-capacitance models that impose static couplings regardless of device state, QDFlow ties these parameters directly to the underlying physics through its self-consistent charge density, producing capacitances and observables that evolve dynamically with gate voltages. This distinction not only improves the connection with the experiment but also allows for the capture of nontrivial behaviors—such as QDs merging, fluctuating slopes, and disorder-induced effects—that are inaccessible to static-capacitance approaches.

As QD systems advance toward larger arrays and integration into functional quantum processors, the need for such realism will only grow. QDFlow package is currently being used to generate datasets for multiple ongoing research projects and we intend to support it for the foreseeable future. Although the current release is stable, we anticipate adding further modules and functionality to the package. Possible future extensions of QDFlow, including

multi-dimensional modeling, adding additional quantum effects, hybridization with experimental feedback loops, and systematic studies of robustness under different noise and disorder regimes, could establish it as a cornerstone for bridging theory, experiment, and ML in the quest for scalable quantum technologies. We have designed the core class structure to allow for seamless addition of new physical parameters and functions, and plan to update the `generate` module to provide support for parallelization. By releasing QDFlow as an open-source package, we aim to foster a shared foundation for accelerating progress in quantum dot technologies. We anticipate that the package will not only continue to advance automated QD control but also provide a flexible testbed for exploring broader questions at the intersection of condensed matter physics, quantum information, and machine learning.

# Acknowledgements

**Funding information**   D.B. was supported in part by an ARO grant no. W911NF-24-2-0043. S.S.K. acknowledges financial support from the S.N. Bose Fellowship during this project. This research was performed in part while J.Z. held an NRC Research Associateship award at NIST. The views and conclusions contained in this paper are those of the authors and should not be interpreted as representing the official policies, either expressed or implied, of the U.S. Government. The U.S. Government is authorized to reproduce and distribute reprints for Government purposes, notwithstanding any copyright noted herein. Any mention of commercial products is for information only; it does not imply recommendation or endorsement by NIST.

**Code availability**   QDFlow is available on the Python Package Index, with the source code released under the GNU General Public License, and can be installed using `pip install QDFlow` command. The associated GitHub repository is https://github.com/QDFlow/QDFlow-solver. Documentation is available at https://qdflow-sim.readthedocs.io/. Any discovered bugs should be reported using GitHub issues. If you find this package useful, please star the repository and cite this paper.

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
