# Peer review of "QDFlow: A Python package for physics simulations of quantum dot devices"

_SciPost Physics Codebases_

## Round 1 · Referee Report · Anonymous (Referee 1) · 2025-10-12

Disclosure of Generative AI use

The referee discloses that the following generative AI tools have been used in the preparation of this report:

GitHub Copilot 1.7.4421 to correct grammar and typos.
Gemini 2.5 flash for brainstorming BEFORE writing a review.

Strengths

1- Proven research impact 2- Novel physics-informed method 3- Simplicity in comparison to previous methods 4- Flexible noise parameters 5-Utility in machine learning pipeline

Weaknesses

1-Heuristic model of quantum dots
2-Uncertain application to heterostructure devices
3-Difficulty in scaling up
4-Lack of benchmark against real charge stability diagram or another simulator
5-Lack of quantum effects, like temperature, spin

Report

The authors present a novel approach to simulating a charge stability diagram, which is a key tool in tuning spin qubit devices. Their approach is based on a Thomas-Fermi simulator, which solves the self-consistent equations for the charge density and potential in the device. The authors explain the problem in convergence and present a method to solve it. Next, they introduce a noise mode, which includes a dozen realistic noise types that can be added to the simulated data, bridging the gap to experimental data. Crucially, the package avoids the constant-capacitance model, or relying on pre-defined capacitance matrices which are difficult to obtain experimentally. Instead, it works directly with the charge density, derived from system geometry, voltages, and material parameters. I find this approach refreshing, novel, and relevant for the community, as it allows for the generation of large, realistic, and accurately labeled datasets for training machine learning models, which is a known bottleneck for training data-based tuning algorithms.

Nevertheless, the package raises some concerns regarding its scalability, utility for non-nanowire devices, and the reality of the model. The package explicitly targets 1D devices, and despite a single positive example in a cited publication, its application to modern 2D dot arrays, or in semiconductor heterostructures, is uncertain. The heuristics used in the pipeline, like the thresholding or definitions of single-/double-dot, are justified but lack clear validation. The paper lacks benchmarks against other models, or even against experimental data, which would help to assess the quality of the simulations. The package also misses documentation, which prevented me from testing it beyond the examples provided in the paper and in the GitHub tests folder. In the longer perspective, besides tunnel coupling, I see no simple way to involve other quantum effects like valley splitting or spin physics, which are crucial for tuning spin qubits. Finally, some concerns arise regarding scalability. The package correctly identifies the difficulty in solving the integer optimization problem, which could become intractable beyond the double quantum dot presented in the paper.

Overall, I believe the package is an important step forward in simulating and training autonomous tuning algorithms needed for further progress in spin qubits. However, the paper needs to be improved in terms of clarity, justification, and benchmarking. I recommend a major revision in response to the points listed below.

Requested changes

1- Before publication, provide a documentation website, or if documentation already exists, place it in visible location on GitHub or pypi page. The documentation should include installation instructions, examples and API reference. 2- Provide benchmarks of the package, including the time needed to generate a single CSD, and how it scales with the system size (number of dots). 3- Provide validation of the model, either by comparing to experimental data, or to other simulators. This could include qualitative comparison of the CSDs, or quantitative comparison of the capacitance matrices. 4- Include a discussion on the limitations of the model, particularly in relation to its applicability to 2D systems and other quantum effects not currently considered. 5- Discuss the roadmap for future developement, particularly in relation to scalability (optimization methods, polytope finding etc.) and possibility of including other quantum effects (valley, spin etc.). Is the package paralleizable? Can it have a learnable parameters to improve the speed/accuracy trade-off?

Recommendation

Ask for minor revision

  • validity: high
  • significance: top
  • originality: top
  • clarity: good
  • formatting: good
  • grammar: excellent

Author:  Justyna Zwolak  on 2025-12-01  [id 6091]

(in reply to Report 1 on 2025-10-12)
Disclosure of Generative AI use

The comment author discloses that the following generative AI tools have been used in the preparation of this comment:

Grammarly Pro (1.144.1.0) to correct grammar and typos.

Category:
answer to question
reply to objection

We thank the referee for their time and work in reviewing our manuscript. In the following, we address each issue raised in their reports point by point.

Point-by-point responses

The authors present a novel approach to simulating a charge stability diagram, which is a key tool in tuning spin qubit devices. Their approach is based on a Thomas-Fermi simulator, which solves the self-consistent equations for the charge density and potential in the device. The authors explain the problem in convergence and present a method to solve it. Next, they introduce a noise mode, which includes a dozen realistic noise types that can be added to the simulated data, bridging the gap to experimental data. Crucially, the package avoids the constant-capacitance model, or relying on pre-defined capacitance matrices which are difficult to obtain experimentally. Instead, it works directly with the charge density, derived from system geometry, voltages, and material parameters. I find this approach refreshing, novel, and relevant for the community, as it allows for the generation of large, realistic, and accurately labeled datasets for training machine learning models, which is a known bottleneck for training data-based tuning algorithms.

Response: We thank the Referee for their kind words about our work.

Nevertheless, the package raises some concerns regarding its scalability, utility for non-nanowire devices, and the reality of the model. The package explicitly targets 1D devices, and despite a single positive example in a cited publication, its application to modern 2D dot arrays, or in semiconductor heterostructures, is uncertain.

Response: QDFlow was initially developed to model 1D arrays, which is why the nanowire model was chosen. Despite this, we have found ML models trained using data generated with QDFlow to work remarkably well in various scenarios (state recognition, virtualization, spurious dot detection, data quality control), on devices fabricated in a university cleanroom and in an industrial setting, on various types of devices Si/SiGe, Ge/SiGe, GaAs/AlGaAs) and on 1D and 2D arrays. References to all demonstrations are given in the Introduction.

That being said, it is true that there are some effects in 2D arrays that QDFlow is currently unable to reproduce. We hope to extend QDFlow in the future to fully model 2D device geometries. This will require a substantial rework of the first part of the simulation, which models the potential of the gates and calculates the charge density function n(x); however, the rest of the simulation will remain largely unchanged. We are considering adding this functionality in a future update.

The heuristics used in the pipeline, like the thresholding or definitions of single-/double-dot, are justified but lack clear validation.

Response: We have tested state-recognition ML models trained exclusively on QDFlow data to differentiate between left/central/right single-dot and double-dot states experimentally within state-tuning algorithms on Si/SiGe and GaAs/AlGaAs, finding no issues with generalizing to experimental data that captures these states. A more quantitative validation was performed as part of Ref. [25], with manually labeled experimental data of various quality levels.

The paper lacks benchmarks against other models, or even against experimental data, which would help to assess the quality of the simulations.

Response: We have added a section in the paper discussing both benchmarking and the limitations of QDFlow. Specifically, we note that for a low number of dots, the main bottleneck of QDFlow is the calculation of the charge density function $n(x)$. As a result, the runtime of QDFlow is orders of magnitude longer than that of simulations using constant capacitive models.

The upside is that qualitatively, QDFlow is able to produce many of the features observed in experimental data that constant capacitance models cannot, such as transition slopes and charging energies that vary across a single CSD, or the “center-dot” regime with two dots merged together.

The package also misses documentation, which prevented me from testing it beyond the examples provided in the paper and in the GitHub tests folder.

Response: We have compiled the documentation in HTML format, which is publicly available here.

In the longer perspective, besides tunnel coupling, I see no simple way to involve other quantum effects like valley splitting or spin physics, which are crucial for tuning spin qubits.

Response: QDFlow currently employs a semiclassical approximation and therefore does not account for most quantum effects. In principle, quantum effects could be taken into account by using a Hubbard model after identifying the tunnel coupling and capacitive energies of each charge configuration. Unfortunately, solving the Hubbard model is very computationally intensive, so some additional approximations or simplifications may be necessary. We would like to add functionality for quantum effects in a future update, but we need to consider the best approach to implement it.

At the same time, QDFlow was designed to enable the automation of tuning phases where quantum effects are somewhat less important. As multiple experimental demonstrations confirm, these objectives are met by the semiclassical version of the simulator.

Finally, some concerns arise regarding scalability. The package correctly identifies the difficulty in solving the integer optimization problem, which could become intractable beyond the double quantum dot presented in the paper.

Response: We have added a section to the paper benchmarking the runtime of the first part of our simulation, which solves for the charge density $n(x)$, and the second part of our simulation, which solves the integer optimization problem. For less than 20 dots, the main bottleneck is calculating $n(x)$, since the integer optimization problem can be quickly brute-forced for small numbers of dots. For more than 20 dots, the exponential time scaling of the integer optimization problem dominates, and thus QDFlow is limited to systems of around 20 dots. However, this 20-dot limit should be more than sufficient for many applications, as long-range Coulomb interactions tend to be screened out at that range, anyway.

Overall, I believe the package is an important step forward in simulating and training autonomous tuning algorithms needed for further progress in spin qubits. However, the paper needs to be improved in terms of clarity, justification, and benchmarking. I recommend a major revision in response to the points listed below.

Response: In what follows, we explain how we address each request.

1-Before publication, provide a documentation website, or if documentation already exists, place it in visible location on GitHub or pypi page. The documentation should include installation instructions, examples and API reference.

Response: We have compiled the documentation in HTML format, which is publicly available at https://qdflow-sim.readthedocs.io/ and is linked in the paper and the GitHub repository.

2-Provide benchmarks of the package, including the time needed to generate a single CSD, and how it scales with the system size (number of dots).

Response: We have added a section to the paper benchmarking the runtime of QDFlow. Specifically, we compare the runtime of solving for $n(x)$ to the runtime of solving the integer optimization problem, both as a function of system size. We find that solving for $n(x)$ is the main bottleneck until 20 dots, at which point the exponential time required for solving the integer optimization problem begins to dominate.

3-Provide validation of the model, either by comparing to experimental data, or to other simulators. This could include qualitative comparison of the CSDs, or quantitative comparison of the capacitance matrices.

Response: Unfortunately, due to several approximations in our model (such as the 1D nature of the nanowire, simplistic modeling of gates, and semiclassical approximations), data generated by QDFlow is not quantitatively robust. However, multiple specialized ML models trained exclusively on data generated using QDFlow and deployed experimentally on samples fabricated in an academic cleanroom and on an industrial 300 mm process line, using Si/SiGe, Ge/SiGe, and GaAs/AlGaAs, and on 1D and 2D QD arrays, confirm the utility of QDFlow. A more quantitative validation was performed as part of Ref. [25], with manually labeled experimental data of various quality levels. We have added this information in the revised version of the manuscript.

Moreover, qualitatively, QDFlow captures many of the features observed in experimental data that constant capacitance models are unable to reproduce. For example, QDFlow data contains transition slopes and charging energies that vary across a single charge stability diagram, as well as a “center-dot” regime, where two adjacent dots merge together if the barrier is too low. Additionally, we have cited examples of applications of QDFlow datasets in current experimental research, demonstrating that ML models trained on QDFlow data have performed well on experimental data.

4-Include a discussion on the limitations of the model, particularly in relation to its applicability to 2D systems and other quantum effects not currently considered.

Response: We have included a new section in the paper that extensively discusses the limitations of our model and simulation. Specifically, we discuss the 1D nature of our nanowire model, the semiclassical Thomas-Fermi approximation employed by our model, which excludes certain quantum effects captured by, for example, the Hubbard model, and the inability of our simulation to model complex latching effects beyond the simple latching noise covered in the noise module.

5-Discuss the roadmap for future developement, particularly in relation to scalability (optimization methods, polytope finding etc.) and possibility of including other quantum effects (valley, spin etc.). Is the package paralleizable? Can it have a learnable parameters to improve the speed/accuracy trade-off?.

Response: We have added a paragraph discussing planned extensions to QDFlow. In particular, we aim to introduce the option of using a 2D model instead of a nanowire to more faithfully represent experimental device architectures. We have designed the general class structure to allow for the addition of additional physical parameters or functions with minimal disruptions to the current codebase. We may add the option to include additional quantum effects, such as Hubbard model physics – although this may further increase computation time. We are also exploring the incorporation of hybridization with experimental feedback loops.

Generating large datasets with QDFlow should be able to take advantage of parallelization, as individual diagrams can be generated independently from each other. We may update the generate module to provide support for parallelization.

We are attaching the differential file, with the changes marked in blue.

Attachment:

QDFlow__Quantum_dot_simulatorredline.pdf

---

## Round 2 · Referee Report · Anonymous (Referee 1) · 2025-12-22

The referee discloses that the following generative AI tools have been used in the preparation of this report:
Assisted by Gemini 3 Pro (accessed December 21, 2025) for language editing and preliminary verification of the final draft. The author takes full responsibility for the accuracy of the content.
Strengths
1- Proven research impact 2- Novel physics-informed method 3- Simplicity in comparison to previous methods 4- Flexible noise parameters 5- Utility in machine learning pipeline 6- Clear scalability benchmark, 7- Good contextualization
Weaknesses
1- Lack of quantum effects, like temperature, spin, tunnel coupling 2- Heuristic model of quantum dots 3-Uncertain applicability to experimentally relevant 2D quantum dot arrays
Report
I agree with their justification that for training machine learning models for autonomous tuning algorithms, capturing qualitative features (such as transition slopes and charging energies) is more critical than exact quantitative matching of capacitance matrices. The references provided regarding the experimental success of ML models trained on this data support this claim.
However, I stand by my previous comments that further progress in autonomous tuning depends on more accurate modeling of quantum effects, in particular tunnel couplings inducing state hybridization and anticrossings, which would be even more visible in 2D arrays. In those systems, larger crosstalk would most likely break the quasi-1D model and lead to additional charge transitions modifying the regular shapes of Coulomb diamonds. As an example, I am attaching a CSD of a 2x2 dot array. Simultaneously, I recognize the authors' empirical evidence for the model's utility for 2D array systems, and their honest discussion of the limitations in the newly added sentences.
The new benchmarking section clearly identifies the scalability bottlenecks, particularly the limit imposed by the integer optimization problem for arrays larger than ~20 dots. I have verified the documentation website, which now includes a detailed description of relevant functions, but still perhaps misses tutorial-style examples.
Overall, I believe the revised manuscript is much improved and provides a clear picture of the capabilities and limitations of QDFlow. After addressing a few minor comments below, I recommend acceptance of the manuscript in its revised form.
Requested changes
1- Could you discuss how the empirical argument underlying transfer learning holds for 2D arrays in the presence of crosstalk and charge hybridisation? Under what conditions can QDFlow be applied to heterostructure-based dots relevant to the community?
2- Following 1, I suggest adding more explicit references to other simulators or models that have attempted to model relevant quantum effects in 2D structures, for instance tunnel coupling. This would help contextualize the limitations of the current semiclassical approach.
3- I suggest improving the GitHub repository to include dependencies, installation instructions, and an example notebook that walks through generating a charge stability diagram. The notebook should be reflected in the documentation website as well, with a tutorial section.
Recommendation
Publish (easily meets expectations and criteria for this Journal; among top 50%)

---

## Round 2 · Author Response

List of changes
- Added a full section and figure discussing benchmarks, testing, and limitations of QDFlow.
- Added several sentences to the introduction emphasizing prior work validating QDFlow data in experimental contexts.
- Added a link to the QDFlow API documentation in the introduction and in the code availability section.
- Added a sentence in the introduction mentioning the unit and benchmark tests in the QDFlow repository.
- Added a few sentences in the conclusion discussing planned future updates to QDFlow.

---

## Round 2 · List of Changes

- Added a full section and figure discussing benchmarks, testing, and limitations of QDFlow.
- Added several sentences to the introduction emphasizing prior work validating QDFlow data in experimental contexts.
- Added a link to the QDFlow API documentation in the introduction and in the code availability section.
- Added a sentence in the introduction mentioning the unit and benchmark tests in the QDFlow repository.
- Added a few sentences in the conclusion discussing planned future updates to QDFlow.

---

## Editorial Decision

unknown